# H2RBox-v2: Incorporating Symmetry for Boosting Horizontal Box Supervised Oriented Object Detection

**Yi Yu**[1*]**, Xue Yang**[2,3*]**, Qingyun Li**[4]**, Yue Zhou**[2]**, Gefan Zhang**[5]**, Feipeng Da**[1†]**, Junchi Yan**[2,3†]

[1]Southeast University    [2]MoE Key Lab of Artificial Intelligence, Shanghai Jiao Tong University
[3]Shanghai AI Laboratory    [4]Harbin Institute of Technology    [5]COWAROBOT Co. Ltd.
{yuyi,dafp}@seu.edu.cn, 21b905003@stu.hit.edu.cn
{yangxue-2019-sjtu,sjtu_zy,lizaozhouke,yanjunchi}@sjtu.edu.cn
PyTorch Code: https://github.com/open-mmlab/mmrotate

## Abstract

With the rapidly increasing demand for oriented object detection, e.g. in autonomous driving and remote sensing, the recently proposed paradigm involving weakly-supervised detector H2RBox for learning rotated box (RBox) from the more readily-available horizontal box (HBox) has shown promise. This paper presents H2RBox-v2, to further bridge the gap between HBox-supervised and RBox-supervised oriented object detection. Specifically, we propose to leverage the reflection symmetry via flip and rotate consistencies, using a weakly-supervised network branch similar to H2RBox, together with a novel self-supervised branch that learns orientations from the symmetry inherent in visual objects. The detector is further stabilized and enhanced by practical techniques to cope with peripheral issues e.g. angular periodicity. To our best knowledge, H2RBox-v2 is the first symmetry-aware self-supervised paradigm for oriented object detection. In particular, our method shows less susceptibility to low-quality annotation and insufficient training data compared to H2RBox. Specifically, H2RBox-v2 achieves very close performance to a rotation annotation trained counterpart – Rotated FCOS: 1) DOTA-v1.0/1.5/2.0: 72.31%/64.76%/50.33% vs. 72.44%/64.53%/51.77%; 2) HRSC: 89.66% vs. 88.99%; 3) FAIR1M: 42.27% vs. 41.25%.

## 1 Introduction

Object detection has been studied extensively, with early research focusing mainly on horizontal detection [1, 2]. When fine-grained bounding boxes are required, oriented object detection [3] is considered more preferable, especially in complex scenes such as aerial images [4–8], scene text [9–13], retail scenes [14], and industrial inspection [15, 16].

With oriented object detection being featured, some horizontal-labeled datasets have been re-annotated, such as DIOR [17] to DIOR-R [18] for aerial image (192K instances) and SKU110K [19] to SKU110K-R [14] for retail scene (1,733K instances). Although re-annotation enables the training of oriented detectors, two facts cannot be ignored: **1)** Horizontal boxes (HBoxes) are more readily available in existing datasets; **2)** Rotated box (RBox) or Mask annotation are more expensive.

---

[*]Equal contribution.

[†]Corresponding author. The work was partly supported by National Natural Science Foundation of China (62306069, 62222607), Shanghai Municipal Science and Technology Major Project (2021SHZDZX0102), Special Project on Basic Research of Frontier Leading Technology of Jiangsu Province of China (BK20192004C), and China Postdoctoral Science Foundation (2023M740602). Yi Yu is also supported by Jiangsu Funding Program for Excellent Postdoctoral Talent (2023ZB616).

37th Conference on Neural Information Processing Systems (NeurIPS 2023).

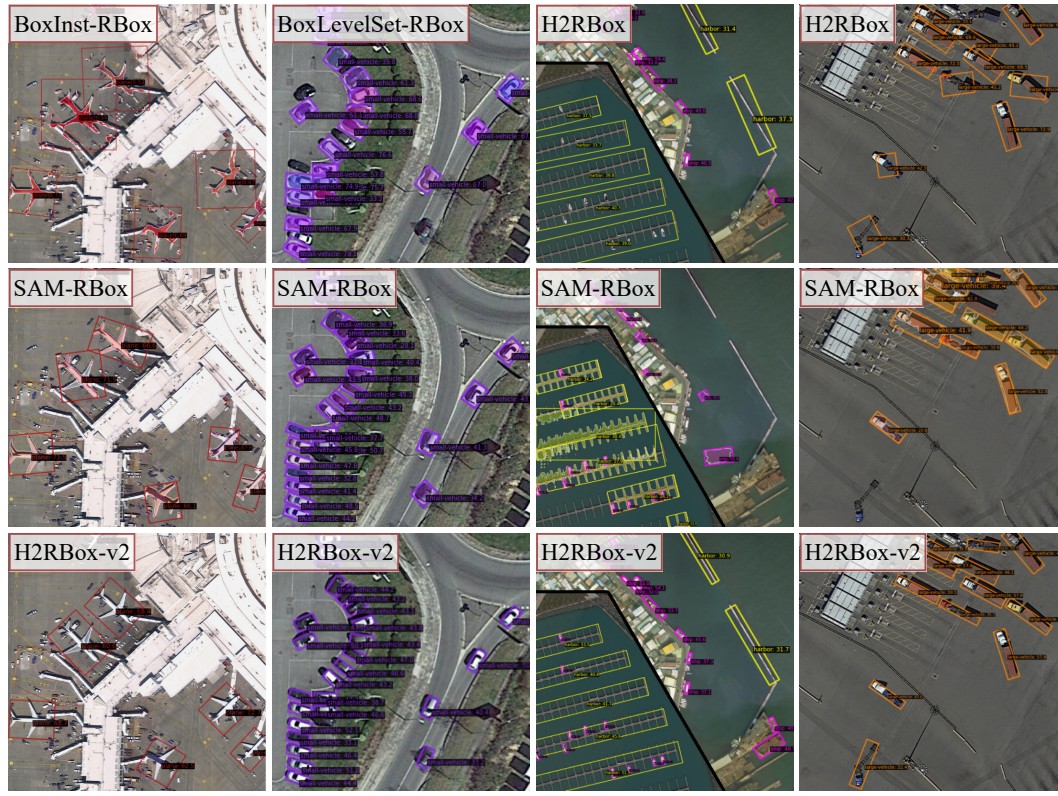

Figure 1: Visual comparisons of HBox-supervised oriented detectors, including BoxInst-RBox (2021) [21], BoxLevelSet-RBox (2022) [22], H2RBox (2023) [20] in the first row, SAM-RBox (2023) [23] in the second row, and H2RBox-v2 (our work) in the third row.

Such a situation raises an interesting question: Can we achieve oriented object detection directly under the weak supervision of HBox annotation? Yang et al. [20] explored this new task setting and proposed the first general solution called HBox-to-RBox (H2RBox) in 2023.

H2RBox gives an effective paradigm and outperforms potential alternatives, including HBox-Mask-RBox (generating RBoxes from segmentation mask) powered by BoxInst [21] and BoxLevelSet [22], the state-of-the-art HBox-supervised instance segmentation methods. Yet it is not impeccable at least in two folds: **1)** H2RBox learns the angle from the geometry of circumscribed boxes, requiring high annotation quality and a large number of training data containing the same object in various orientations. **2)** H2RBox requires quite restrictive working conditions. For example, it is incompatible with rotation augmentation and is sensitive to black borders on images.

If the above requirements are not met, H2RBox may perform far below expectations. As a result, H2RBox does not support some small datasets (e.g. HRSC [5]) and some augmentation methods (e.g. random rotation), leading to bottlenecks in its application scenarios and performance.

By rethinking HBox-supervised oriented object detection with a powerful yet unexplored theory, symmetry-aware self-supervision, we present H2RBox-v2, a new version of H2RBox that exploits the symmetry of objects and solves the above issues.

**Motivation:** Symmetry is a natural and popular property in vision [24]. For example, in the DOTA dataset [4], many categories (planes, courts, vehicles, ships, etc.) show significant reflection symmetry. For RBox annotation, symmetry is also an important consideration—Humans can intuitively take the direction of the symmetrical axis as the orientation, even without annotation guidance. Is there a way to use symmetry as a supervising tool in deep neural networks for oriented object detection? Will this technique lead to better performance? These are the questions for which this paper is intended.

**What is new in H2RBox-v2?  1)** A new self-supervised branch leaning angle from symmetry. H2RBox-v2 directly learns angles from the image through the symmetry of objects. That means it is capable of learning the angles correctly even if the HBox annotations are inaccurate in size (not

Table 1: Comparison between H2RBox [20] (denoted as "v1") and H2RBox-v2 (ours).

| | Arch.[1] | SS View Transform | Angle Acquisition Principle | Aug. Applicability | | Dataset Applicability[2] | | | Gap[3] |
|---|---|---|---|---|---|---|---|---|---|
| | | | | MS | RR | DOTA | HRSC | FAIR1M | |
| v1 | WS+SS | Rotate | Geometric Constraints Rotate Consistency | ✓ | × | ✓ | × | ✓ | -3.41% |
| v2 | WS+SS | Flip Rotate | Symmetry-aware Learning | ✓ | ✓ | ✓ | ✓ | ✓ | **+0.07%** |

[1] WS: Weakly-supervised, SS: Self-supervised. [2] Based on whether the training converges on the dataset.
[3] Performance gap of $AP_{50}$ compared to RBox-supervision (i.e. FCOS) average on datasets with "✓".

precisely the circumscribed rectangle, see Table 9) or when the training data is relatively insufficient (see Table 10). **2)** A newly designed CircumIoU loss in the weakly-supervised branch. With this amendment, H2RBox-v2 is now compatible with random rotation augmentation. **3)** As a result, H2RBox-v2 gives a higher performance (as is displayed in Fig. 1 and Table 1), further bridging the gap between HBox-supervised and RBox-supervised oriented object detection.

**Contributions: 1)** This work is the first attempt to explore symmetry for angle regression in oriented object detection, showing that symmetry in images can help learn the orientation of objects in a self-supervised manner. **2)** A new training paradigm incorporating symmetry in images is elaborated on, and an integral and stable implementation is provided in this paper. **3)** The proposed methods are evaluated via extensive experiments, showing their capability of learning the orientation via symmetry, achieving a higher accuracy than H2RBox. The source code is publicly available.

## 2 Related Work

**RBox-supervised oriented object detection:** Representative works include anchor-based detector Rotated RetinaNet [25], anchor-free detector Rotated FCOS [26], and two-stage detectors such as RoI Transformer [27], Oriented R-CNN [28] and ReDet [29]. Besides, $R^3$Det [30] and $S^2$A-Net [31] improve the performance by exploiting alignment features. Most of the above methods directly perform angle regression, which may face loss discontinuity and regression inconsistency induced by the periodicity of the angle. Remedies have been thus developed, including modulated losses [32, 33] that alleviate loss jumps, angle coders [34–36] that convert the angle into boundary-free coded data, and Gaussian-based losses [37–40] that transform rotated bounding boxes into Gaussian distributions. Additionally, RepPoint-based methods [41–43] provide new alternatives for oriented object detection, which predict a set of sample points that bounds the spatial extent of an object.

**HBox-supervised instance segmentation:** Compared with HBox-supervised oriented object detection, HBox-supervised instance segmentation, a similar task also belonging to weakly-supervised learning, has been better studied in the literature. For instance, SDI [44] refines the segmentation through an iterative training process; BBTP [45] formulates the HBox-supervised instance segmentation into a multiple-instance learning problem based on Mask R-CNN [46]; BoxInst [21] uses the color-pairwise affinity with box constraint under an efficient RoI-free CondInst [47]; BoxLevelSet [22] introduces an energy function to predict the instance-aware mask as the level set; SAM (Segment Anything Model) [23] produces object masks from input prompts such as points or HBoxes.

Most importantly, these HBox-supervised instance segmentation methods are potentially applicable to HBox-supervised oriented object detection by finding the minimum circumscribed rectangle of the segmentation mask. Such an HBox-Mask-RBox paradigm is a potential alternative for the task we are aiming at and is thus added to our experiments for comparison.

**HBox-supervised oriented object detection:** Though the oriented bounding box can be obtained from the segmentation mask, such an HBox-Mask-RBox pipeline can be less cost-effective. The seminal work H2RBox [20] circumvents the segmentation step and achieves RBox detection directly from HBox annotation. With HBox annotations for the same object in various orientations, the geometric constraint limits the object to a few candidate angles. Supplemented with a self-supervised branch eliminating the undesired results, an HBox-to-RBox paradigm is established.

Some similar studies use additional annotated data for training, which are also attractive but less general than H2RBox: **1)** OAOD [48] is proposed for weakly-supervised oriented object detection. But in fact, it uses HBox along with an object angle as annotation, which is just "slightly weaker" than RBox supervision. Such an annotation manner is not common, and OAOD is only verified

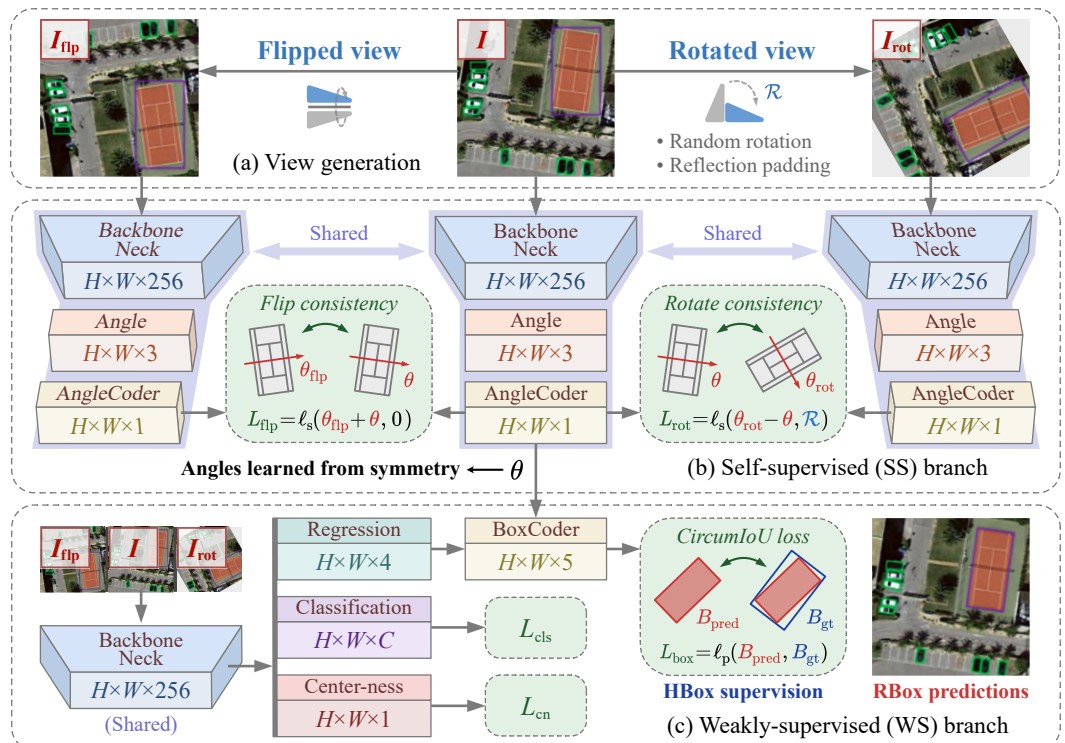

Figure 2: The overview of H2RBox-v2, consisting of a self-supervised branch that learns angles from the symmetry of objects, and a weakly-supervised branch that learns other properties from HBoxes.

on their self-collected ITU Firearm dataset. **2)** Sun et al. [49] proposes a two-stage framework: i) training detector with the annotated horizontal and vertical objects. ii) mining the rotation objects by rotating the training image to align the oriented objects as horizontally or vertically as possible. **3)** KCR [50] combines RBox-annotated source datasets with HBox-annotated target datasets, and achieves HBox-supervised oriented detection on the target datasets via transfer learning.

**Symmetry detection:** Detecting the symmetry (e.g. reflection) has been a long-standing research topic in vision [24], with approaches including key point matching [51, 52], iterative optimization [53, 54], and deep learning [55, 56]. These methods are aimed at tasks quite different from this paper, and we are the first to introduce symmetry-aware learning for oriented object detection.

## 3 Proposed Method

An overview of the proposed H2RBox-v2 is given in Fig. 2, which consists of a self-supervised (SS) branch (Sec. 3.2) and a weakly-supervised (WS) branch (Sec. 3.3).

### 3.1 Preliminaries and Approach Overview

In H2RBox-v2, SS branch is designed to learn the orientation of objects from symmetry via consistencies between several views of training images, whilst WS branch learns other properties from the HBox annotation. The core idea of symmetry-aware SS learning is described below.

**Definition of reflection symmetry:** An object has reflectional symmetry if there is a line going through it which divides it into two pieces that are mirror images of each other [57].

Assume there is a neural network $f_{\text{nn}}(\cdot)$ that maps a symmetric image $I$ to a real number $\theta$:

$$\theta = f_{\text{nn}}(I) \tag{1}$$

To exploit reflection symmetry, we endow the function with two new properties: flip consistency and rotate consistency.

**Property I: Flip consistency.** With an input image vertically flipped, $f_{\text{nn}}(\cdot)$ gives an opposite output:

$$f_{\text{nn}}(I) + f_{\text{nn}}(\text{flp}(I)) = 0 \tag{2}$$

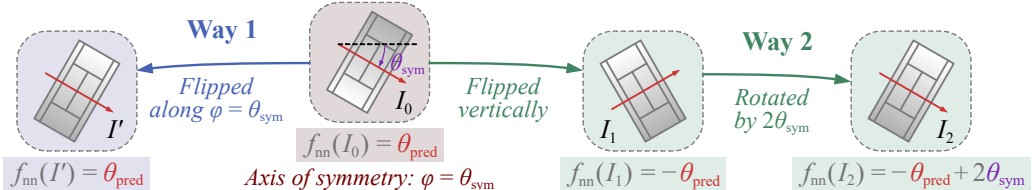

Figure 3: To illustrate that the reflection symmetry (Way 1) is equivalent to a vertical flip plus a rotation (Way 2). The expected outputs at the bottom show that $f_{nn}(I') = f_{nn}(I_2) \Rightarrow \theta_{pred} = \theta_{sym}$.

where $\text{flp}(I)$ is an operator of vertically flipping the image $I$.

**Property II: Rotate consistency.** With an input rotated by $\mathcal{R}$, the output of $f_{nn}(\cdot)$ also rotates by $\mathcal{R}$:

$$f_{nn}(\text{rot}(I, \mathcal{R})) - f_{nn}(I) = \mathcal{R} \tag{3}$$

where $\text{rot}(I, \mathcal{R})$ is an operator that clockwise rotates the image $I$ by $\mathcal{R}$.

Now we consider that given an image $I_0$ symmetric about $\varphi = \theta_{sym}$, assuming the corresponding output is $\theta_{pred} = f_{nn}(I_0)$, the image can be transformed in two ways as shown in Fig. 3:

• **Way 1**: Flipping $I_0$ along line $\varphi = \theta_{sym}$. According to the above definition of reflection symmetry, the output remains the same, i.e. $f_{nn}(I') = f_{nn}(I_0) = \theta_{pred}$.

• **Way 2**: Flipping $I_0$ vertically to obtain $I_1$ first, and then rotating $I_1$ by $2\theta_{sym}$ to obtain $I_2$. According to flip and rotate consistencies, the output is supposed to be $f_{nn}(I_2) = -\theta_{pred} + 2\theta_{sym}$.

On the ground that ways 1 and 2 are equivalent, the transformed images $I'$ and $I_2$ are identical. And thus $f_{nn}(I') = f_{nn}(I_2)$, finally leading to $\theta_{pred} = \theta_{sym}$.

From the above analysis, it arrives at a conclusion that if image $I_0$ is symmetric about line $\varphi = \theta_{sym}$ and function $f_{nn}(\cdot)$ subjects to both flip and rotate consistencies, then $f_{nn}(I_0)$ must be equal to $\theta_{sym}$, so the orientation is obtained. In the following, we further devise two improvements.

**Handling the angle periodicity:** For higher conciseness, the periodicity is not included in the above formula. To allow images to be rotated into another cycle, the consistencies are modified as:

$$\begin{aligned} f_{nn}(I) + f_{nn}(\text{flp}(I)) &= k\pi \\ f_{nn}(\text{rot}(I, \mathcal{R})) - f_{nn}(I) &= \mathcal{R} + k\pi \end{aligned} \tag{4}$$

where $k$ is an integer to keep left and right in the same cycle. This problem is coped with using the snap loss in Sec. 3.4. Meanwhile, the conclusion should be amended as: $f_{nn}(I_0) = \theta_{sym} + k\pi/2$, meaning that the network outputs either the axis of symmetry or a perpendicular one.

**Extending to the general case of multiple objects:** Strictly speaking, our above discussion is limited to the setting when image $I_0$ contains only one symmetric object. In detail implementation, we use an assigner to match the center of objects in different views, and the consistency losses are calculated between these matched center points. We empirically show on several representative datasets (see Sec. 4.2) that our above design is applicable for multiple object detection with objects not perfectly symmetric, where an approximate axis of each object can be found via learning.

## 3.2 Self-supervised (SS) Branch

The above analysis suggests that the network can learn the angle of objects from symmetry through the flip and rotate consistencies. A self-supervised branch is accordingly designed.

During the training process, we perform vertical flip and random rotation to generate two transformed views, $I_{flp}$ and $I_{rot}$, of the input image $I$, as shown in Fig. 2 (a). The blank border area induced by rotation is filled with reflection padding. Afterward, the three views are fed into three parameter-shared branches of the network, where ResNet50 [58] and FPN [59] are used as the backbone and the neck, respectively. The random rotation is in range $\pi/4 \sim 3\pi/4$ (according to Table 7).

Similar to H2RBox, a label assigner is required in the SS branch to match the objects in different views. In H2RBox-v2, we calculate the average angle features on all sample points for each object and eliminate those objects without correspondence (some objects may be lost during rotation).

Following the assigner, an angle coder PSC [36] is further adopted to cope with the boundary problem. Table 4 empirically shows that the impact of boundary problem in our self-supervised setting could

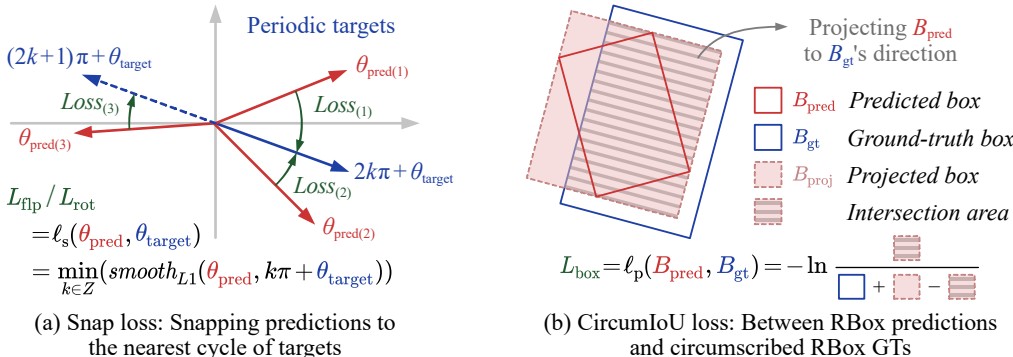

(a) Snap loss: Snapping predictions to the nearest cycle of targets

(b) CircumIoU loss: Between RBox predictions and circumscribed RBox GTs

Figure 4: Illustration of the snap loss (for SS branch) and the CircumIoU loss (for WS branch).

be much greater than that in the supervised case, especially in terms of stability. The decoded angles of the original, flipped, and rotated views are denoted as $\theta$, $\theta_{\text{flp}}$, and $\theta_{\text{rot}}$.

Finally, with formula in Sec. 3.4, the loss $L_{\text{flp}}$ can be calculated between $\theta$ and $\theta_{\text{flp}}$, whereas $L_{\text{rot}}$ between $\theta$ and $\theta_{\text{rot}}$. By minimizing $L_{\text{flp}}$ and $L_{\text{rot}}$, the network learns to conform with flip and rotate consistencies and gains the ability of angle prediction through self-supervision.

### 3.3 Weakly-supervised (WS) Branch

The SS branch above provides the angle of objects. To predict other properties of the bounding box (position, size, category, etc.), a weakly-supervised branch using HBox supervision is further introduced. The WS branch in H2RBox-v2 is inherited from H2RBox, but has some differences:

**1)** In H2RBox, the angle is primarily learned by the WS branch, with an SS branch eliminating the undesired results. Comparably, H2RBox-v2 has a powerful SS branch driven by symmetry that can learn the angle independently, and thus the angle subnet is deleted from the WS branch.

**2)** H2RBox converts $B_{\text{pred}}$ to an HBox to calculate the IoU loss [60] with the HBox $B_{\text{gt}}$, which cannot work with random rotation where $B_{\text{gt}}$ becomes an RBox. To solve this problem, CircumIoU loss (detailed in Sec. 3.4) is proposed in H2RBox-v2 to directly calculate $L_{\text{box}}$ between $B_{\text{pred}}$ and $B_{\text{gt}}$, so that $B_{\text{gt}}$ is allowed to be an RBox circumscribed to $B_{\text{pred}}$, as is shown in Fig. 2 (c).

### 3.4 Loss Functions

**Loss for SS branch:** As described in Eq. (4), the consistencies encounter the periodicity problem in rotation. To cope with this problem, the snap loss $\ell_s$ [3] is proposed in this paper:

$$\ell_s\left(\theta_{\text{pred}}, \theta_{\text{target}}\right) = \min_{k \in Z}\left(smooth_{L1}\left(\theta_{\text{pred}}, k\pi + \theta_{\text{target}}\right)\right) \tag{5}$$

where an illustration is displayed in Fig. 4 (a). The snap loss limits the difference between predicted and target angles to $\pi/2$ so that the loss calculation upon the two decoded angles will not induce boundary discontinuity. The snap loss is used in both $L_{\text{flp}}$ and $L_{\text{rot}}$ as:

$$\begin{aligned} L_{\text{flp}} &= \ell_{\text{s}}\left(\theta_{\text{flp}} + \theta, 0\right) \\ L_{\text{rot}} &= \ell_{\text{s}}\left(\theta_{\text{rot}} - \theta, \mathcal{R}\right) \end{aligned} \tag{6}$$

where $L_{\text{flp}}$ is the loss for flip consistency, and $L_{\text{rot}}$ for rotate consistency. $\theta$, $\theta_{\text{flp}}$, and $\theta_{\text{rot}}$ are the outputs of the three views described in Sec. 3.2. $\mathcal{R}$ is the angle in generating rotated view.

With the above definitions, the loss of the SS branch can be expressed as:

$$L_{\text{ss}} = \lambda L_{\text{flp}} + L_{\text{rot}} \tag{7}$$

where $\lambda$ is the weight set to 0.05 according to the ablation study in Table 6.

**Loss for the WS branch:** The losses in the WS branch are mainly defined by the backbone FCOS detector, including $L_{\text{cls}}$ for classification and $L_{\text{cn}}$ for center-ness. Different from RBox-supervised

---

[3]There are a series of targets with interval $\pi$, just like a series of evenly spaced grids. The snap loss moves prediction toward the closest target, thus deriving its name from "snap to grid" function.

methods, $B_{\text{gt}}$ in our task is an RBox circumscribed to $B_{\text{pred}}$, so we re-define the loss for box regression $L_{\text{box}}$ with CircumIoU loss as:

$$\ell_p\left(B_{\text{pred}}, B_{\text{gt}}\right) = -\ln \frac{B_{\text{proj}} \cap B_{\text{gt}}}{B_{\text{proj}} \cup B_{\text{gt}}} \tag{8}$$

where $B_{\text{proj}}$ is the dashed box in Fig. 4 (b), obtained by projecting the predicted box $B_{\text{pred}}$ to the direction of ground-truth box $B_{\text{gt}}$.

The CircumIoU loss enables H2RBox-v2 to use random rotation augmentation to further improve the performance (see Table 2), which is not supported by H2RBox.

Finally, the loss of the WS branch can be expressed as:

$$L_{\text{ws}} = L_{\text{cls}} + \mu_{\text{cn}} L_{\text{cn}} + \mu_{\text{box}} L_{\text{box}} \tag{9}$$

where the hyper-parameters are set to $\mu_{\text{cn}} = 1$ and $\mu_{\text{box}} = 1$ by default.

**Overall loss:** The overall loss of the proposed network is the sum of the WS loss and the SS loss:

$$L_{\text{total}} = L_{\text{ws}} + \mu_{\text{ss}} L_{\text{ss}} \tag{10}$$

where $\mu_{\text{ss}}$ is the weight of SS branch set to 1 by default.

### 3.5 Inference Phase

Although using a multi-branch paradigm in training (as shown in Fig. 2), H2RBox-v2 does not require the multi-branch or the view generation during inference.

Due to the parameter sharing, the inference only involves the forward propagation of backbone, angle head (from SS), and other heads (i.e. regression, classification, and center-ness from WS). The only additional cost of H2RBox-v2 during inference is the PSC decoding (compared to H2RBox). Thus, FCOS/H2RBox/H2RBox-v2 have similar inference speed (see Table 2).

## 4 Experiments

Using PyTorch 1.13.1 [61] and the rotation detection tool kits: MMRotate 1.0.0 [62], experiments are carried out. The performance comparisons are obtained by using the same platforms (i.e. PyTorch/MMRotate version) and hyper-parameters (learning rate, batch size, optimizer, etc.).

### 4.1 Datasets and Settings

**DOTA [4]:** DOTA-v1.0 contains 2,806 aerial images—1,411 for training, 937 for validation, and 458 for testing, as annotated using 15 categories with 188,282 instances in total. DOTA-v1.5/2.0 are the extended version of v1.0. We follow the default preprocessing in MMRotate: The high-resolution images are split into $1,024 \times 1,024$ patches with an overlap of 200 pixels for training, and the detection results of all patches are merged to evaluate the performance.

**HRSC [5]:** It contains ship instances both on the sea and inshore, with arbitrary orientations. The training, validation, and testing set includes 436, 181, and 444 images, respectively. With preprocessing by MMRotate, images are scaled to $800 \times 800$ for training/testing.

**FAIR1M [66]:** It contains more than 1 million instances and more than 40,000 images for fine-grained object recognition in high-resolution remote sensing imagery. The dataset is annotated with five categories and 37 fine-grained subcategories. We split the images into $1,024 \times 1,024$ patches with an overlap of 200 pixels and a scale rate of 1.5 and merge the results for testing. The performance is evaluated on the FAIR1M-1.0 server.

**Experimental settings:** We adopt the FCOS [26] detector with ResNet50 [58] backbone and FPN [59] neck as the baseline method, based on which we develop our H2RBox-v2. We choose average precision (AP) as the primary metric to compare with existing literature. For a fair comparison, all the listed models are configured based on ResNet50 [58] backbone and trained on NVIDIA RTX3090/4090 GPUs. All models are trained with AdamW [67], with an initial learning rate of 5e-5 and a mini-batch size of 2. Besides, we adopt a learning rate warm-up for 500 iterations, and the learning rate is divided by ten at each decay step. "1x", "3x", and "6x" schedules indicate 12, 36, and 72 epochs for training. "MS" and "RR" denote multi-scale technique [62] and random rotation augmentation. Unless otherwise specified, "6x" is used for HRSC and "1x" for the other datasets, while random flipping is the only augmentation that is always adopted by default.

Table 2: Results on the DOTA-v1.0 dataset.

| | Method | Sched. | MS | RR | Size | FPS | AP$_{50}$ |
|---|---|---|---|---|---|---|---|
| RBox-supervised | RepPoints (2019) [41] | 1x | | | 1,024 | 24.5 | 68.45 |
| | RetinaNet (2017) [25] | 1x | | | 1,024 | 25.4 | 68.69 |
| | KLD (2021) [38] | 1x | | | 1,024 | 25.4 | 71.24 |
| | KFIoU (2023) [40] | 1x | | | 1,024 | 25.4 | 71.61 |
| | GWD (2021) [37] | 1x | | | 1,024 | 25.4 | 71.66 |
| | PSC (2023) [36] | 1x | | | 1,024 | 25.4 | 71.92 |
| | SASM (2022) [63] | 1x | | | 1,024 | 24.4 | 72.30 |
| | R$^3$Det (2021) [30] | 1x | | | 1,024 | 20.0 | 73.12 |
| | CFA (2021) [64] | 1x | | | 1,024 | 24.5 | 73.84 |
| | Oriented RepPoints (2022) [43] | 1x | | | 1,024 | 24.5 | 75.26 |
| | S$^2$A-Net (2022) [31] | 1x | | | 1,024 | 23.3 | 75.81 |
| | FCOS (2019) [26] | 1x | | | 1,024 | 29.5 | 72.44 |
| | FCOS (2019) [26] | 3x | | ✓ | 1,024 | 29.5 | 74.75 |
| | FCOS (2019) [26] | 1x | ✓ | ✓ | 1,024 | 29.5 | 77.68 |
| HBox-supervised | BoxInst-RBox (2021) [21] [1] | 1x | | | 960 | 2.7 | 53.59 |
| | BoxLevelSet-RBox (2022) [22] [2] | 1x | | | 960 | 4.7 | 56.44 |
| | SAM-ViT-B-RBox (2023) [23] [3] | 1x | | | 1,024 | 1.7 | 63.94 |
| | H2RBox (FCOS-based) (2023) [20] [4] | 1x | | | 1,024 | 29.1 | 67.82 |
| | H2RBox (FCOS-based) (2023) [20] [5] | 1x | | | 1,024 | 29.1 | 70.05 |
| | H2RBox (FCOS-based) (2023) [20] [4] | 1x | ✓ | | 1,024 | 29.1 | 74.40 |
| | H2RBox (FCOS-based) (2023) [20] [5] | 1x | ✓ | | 1,024 | 29.1 | 75.35 |
| | H2RBox-v2 (FCOS-based) | 1x | | | 960 | 31.6 | 71.46 |
| | H2RBox-v2 (FCOS-based) | 1x | | | 1,024 | 29.1 | 72.31 |
| | H2RBox-v2 (FCOS-based) | 3x | | ✓ | 1,024 | 29.1 | 74.29 |
| | H2RBox-v2 (FCOS-based) | 1x | ✓ | | 1,024 | 29.1 | 77.97 |
| | H2RBox-v2 (FCOS-based) | 1x | ✓ | ✓ | 1,024 | 29.1 | 78.25 |
| | H2RBox-v2 (FCOS-based, Swin-T) [6] | 1x | ✓ | ✓ | 1,024 | 24.0 | 79.39 |
| | H2RBox-v2 (FCOS-based, Swin-B) [6] | 1x | ✓ | ✓ | 1,024 | 12.4 | 80.61 |

[1] "-RBox" means the minimum rectangle operation is performed on the Mask to obtain RBox.

[2] Evaluated on NVIDIA V100 GPU due to the excessive RAM usage.

[3] The code is available at https://github.com/Li-Qingyun/sam-mmrotate.

[4] Results reported in the original paper [20].

[5] Results reproduced by us with same infrastructure for a fair comparison.

[6] Using Swin Transformer [65] as backbone on four NVIDIA A100 GPUs with batch size = 4.

## 4.2 Main Results

**DOTA-v1.0:** Table 2 shows that H2RBox-v2 outperforms HBox-Mask-Rbox methods in both accuracy and speed. Taking BoxLevelSet-RBox [22] as an example, H2RBox-v2 gives an accuracy of 15.02% higher, and a x7 speed faster by avoiding the time-consuming post-processing (i.e. minimum circumscribed rectangle operation). In particular, the recent foundation model for segmentation i.e. SAM [23] has shown strong zero-shot capabilities by training on the largest segmentation dataset to date. Thus, we use a trained horizontal FCOS detector to provide HBoxes into SAM as prompts, so that the corresponding masks can be generated by zero-shot, and finally the rotated RBoxes are obtained by performing the minimum circumscribed rectangle operation on the predicted Masks. Thanks to the powerful zero-shot capability, SAM-RBox based on ViT-B [68] in Table 2 has achieved 63.94%. However, it is also limited to the additional mask prediction step and the time-consuming post-processing, only 1.7 FPS during inference.

In comparison with the current state-of-the-art method H2RBox, to make it fair, we use the reproduced result of H2RBox, which achieves 70.05%, 2.23% higher than the original paper [20]. In this fair comparison, our method outperforms H2RBox by 2.26% (72.31% vs. 70.05%, both w/o MS). When MS is applied, the improvement is 2.62% (75.35% vs. 77.97%, both w/ MS).

Furthermore, the performance gap between our method and the RBox-supervised FCOS baseline is only 0.13% (w/o MS and RR) and 0.46% (w/ RR). When MS and RR are both applied, our method outperforms RBox-supervised FCOS by 0.57% (78.25% vs. 77.68%), proving that supplemented with symmetry-aware learning, the weakly-supervised learning can achieve performance on a par with the fully-supervised one upon the same backbone neural network. Finally, H2RBox-v2 obtains 80.61% on DOTA-v1.0 by further utilizing a stronger backbone.

Table 3: AP$_{50}$ performance on the DOTA-v1.0/1.5/2.0, HRSC, and FAIR1M datasets.

| Method | DOTA-v1.0 | DOTA-v1.5 | DOTA-v2.0 | HRSC | FAIR1M |
|---|---|---|---|---|---|
| RetinaNet (2017) [25] | 68.69 | 60.57 | 47.00 | 84.49 | 37.67 |
| GWD (2021) [37] | 71.66 | 63.27 | 48.87 | 86.67 | 39.11 |
| S$^2$A-Net (2022) [31] | 75.81 | 66.53 | 52.39 | 90.10 | 42.44 |
| FCOS (2019) [26] | 72.44 | 64.53 | 51.77 | 88.99 | 41.25 |
| Sun et al. (2021) [49][1] | 38.60 | - | - | - | - |
| KCR (2023) [50][2] | - | - | - | 79.10 | - |
| H2RBox (2023) [20] | 70.05 | 61.70 | 48.68 | 7.03 | 35.94 |
| H2RBox-v2 | **72.31** | **64.76** | **50.33** | **89.66** | **42.27** |

[1] Sparse annotation for horizontal/vertical objects. The result is cited from their paper.
[2] Transfer learning from DOTA (RBox) to HRSC (HBox). The result is cited from their paper.

Table 4: Ablation with different SS losses.

| Dataset | PSC | $\ell_s$ | AP | AP$_{50}$ | AP$_{75}$ |
|---|---|---|---|---|---|
| DOTA | | | 24.24 | 52.24 | 19.48 |
| | | ✓ | 0.01 | 0.77 | 0.02 |
| | ✓ | | 10.49 | 27.57 | 6.15 |
| | ✓ | ✓ | **40.69** | **72.31** | **39.49** |
| HRSC | | | 2.25 | 7.83 | 0.62 |
| | | ✓ | 48.95 | 88.52 | 50.03 |
| | ✓ | | 0.31 | 0.88 | 0.13 |
| | ✓ | ✓ | **58.03** | **89.66** | **64.80** |

Table 5: Ablation with different WS losses.

| Dataset | $\ell_p$ | RR | AP | AP$_{50}$ | AP$_{75}$ |
|---|---|---|---|---|---|
| DOTA | | | 39.35 | 71.49 | 37.03 |
| | | ✓ | 11.93 | 29.34 | 7.86 |
| | ✓ | | **40.69** | **72.31** | 39.49 |
| | ✓ | ✓ | 40.17 | 71.79 | **39.77** |
| HRSC | | | 56.20 | 89.58 | 61.84 |
| | | ✓ | 41.10 | 87.19 | 33.97 |
| | ✓ | | 58.03 | **89.66** | 64.80 |
| | ✓ | ✓ | **63.82** | 89.56 | **76.11** |

**DOTA-v1.5/2.0:** As extended versions of DOTA-v1.0, these two datasets are more challenging, while the results present a similar trend. Still, H2RBox-v2 shows considerable advantages over H2RBox, with an improvement of 3.06% on DOTA-v1.5 and 1.65% on DOTA-v2.0. The results on DOTA-v1.5/2.0, HRSC, and FAIR1M are shown in Table 3.

**HRSC:** H2RBox [20] can hardly learn angle information from small datasets like HRSC, resulting in deficient performance. Contrarily, H2RBox-v2 is good at this kind of task, giving a performance comparable to fully-supervised methods. Compared to KCR [50] that uses transfer learning from RBox-supervised DOTA to HBox-supervised HRSC, our method, merely using HBox-supervised HRSC, outperforms KCR by 10.56% (89.66% vs. 79.10%).

**FAIR1M:** This dataset contains a large number of planes, vehicles, and courts, which are more perfectly symmetric than objects like bridges and harbors in DOTA. This may explain the observation that H2RBox-v2, learning from symmetry, outperforms H2RBox by a more considerable margin of 6.33% (42.27% vs. 35.94%). In this case, H2RBox-v2 even performs superior to the fully-supervised FCOS that H2RBox-v2 is based on by 1.02% (42.27% vs. 41.25%).

### 4.3 Ablation Studies

**Loss in SS branch:** Table 4 studies the impact of using the snap loss (see Sec. 3.4) and the angle coder. Column "PSC" indicates using PSC angle coder [36] and "w/o PSC" means the conv layer directly outputs the angle. Column "$\ell_s$" with check mark denotes using snap loss (otherwise using smooth L1 loss). Without these two modules handling boundary discontinuity, we empirically find that the loss could fluctuate in a wide range, even failure in convergence (see the much lower results in Table 4). In comparison, when both PSC and snap loss are used, the training is stable.

**Loss in WS branch:** Table 5 shows that CircumIoU loss with random rotation can further improve the performance, which H2RBox is incapable of. "$\ell_p$" means using CircumIoU loss in Sec. 3.4, and otherwise, IoU loss [60] is used following a conversion from RBox to HBox (see H2RBox [20]).

**Weights between $L_{flip}$ and $L_{rot}$:** Table 6 shows that on both DOTA and HRSC datasets, $\lambda = 0.05$ could be the best choice under AP$_{50}$ metric, whereas $\lambda = 0.1$ under AP$_{75}$. Hence in most experiments, we choose $\lambda = 0.05$, except for Table 7 where $\lambda = 0.1$ is used.

**Range of view generation:** When the rotation angle $\mathcal{R}$ is close to 0, the SS branch could fall into a sick state. This may explain the fluctuation of losses under the random rotation within $-\pi \sim \pi$, leading to training instability. According to Table 7, $\pi/4 \sim 3\pi/4$ is more suitable.

Table 6: Ablation with different weights between flipping and rotating losses defined in Eq. 6.

| Dataset | $\lambda$ | AP | $AP_{50}$ | $AP_{75}$ | Dataset | $\lambda$ | AP | $AP_{50}$ | $AP_{75}$ |
|---|---|---|---|---|---|---|---|---|---|
| DOTA | 0 | 31.60 | 66.37 | 25.03 | HRSC | 0 | 0.06 | 0.32 | 0.00 |
| | 0.01 | 40.43 | 72.26 | 38.55 | | 0.01 | 55.78 | 89.20 | 61.72 |
| | 0.05 | **40.69** | **72.31** | 39.49 | | 0.05 | 58.03 | **89.66** | 64.80 |
| | 0.1 | 40.48 | 71.46 | **39.84** | | 0.1 | **58.22** | 89.45 | **64.99** |
| | 0.5 | 39.94 | 72.26 | 38.16 | | 0.5 | 53.85 | 88.90 | 61.47 |
| | 1.0 | 38.50 | 70.91 | 36.02 | | 1.0 | 1.57 | 6.97 | 0.38 |

Table 7: Ablation with different random ranges in the rotated view generation on HRSC.

| Range | AP | $AP_{50}$ | $AP_{75}$ |
|---|---|---|---|
| $-\pi \sim \pi^*$ | 56.57 | 89.47 | 63.14 |
| $\pi/4 \sim 3\pi/4$ | **58.22** | 89.45 | **64.99** |
| $3\pi/8 \sim 5\pi/8$ | 56.81 | **89.83** | 64.03 |
| $7\pi/16 \sim 9\pi/16$ | 55.56 | 89.40 | 61.28 |

$^*$ Not stable, occasionally be much lower.

Table 8: Ablation with different padding strategies for rotated view generation.

| Dataset | Padding | AP | $AP_{50}$ | $AP_{75}$ |
|---|---|---|---|---|
| DOTA | Zeros | 40.49 | 72.26 | 39.15 |
| | Reflection | **40.69** | **72.31** | **39.49** |
| HRSC | Zeros | 55.90 | 89.32 | 60.95 |
| | Reflection | **58.03** | **89.66** | **64.80** |

Table 9: Ablation with different levels of noise adding to HBox annotations on DOTA.

| $\sigma$ | H2RBox | | H2RBox-v2 | |
|---|---|---|---|---|
| | $AP_{50}$ | $AP_{75}$ | $AP_{50}$ | $AP_{75}$ |
| 0% | 70.05 | 38.38 | 72.31 | 39.49 |
| 10% | 69.19 | 35.24 | 71.68 | 36.33 |
| 30% | 67.39 | 26.02 | 71.11 | 34.12 |

Table 10: Ablation training with different sampling percentages of DOTA dataset.

| $p$ | H2RBox | | H2RBox-v2 | |
|---|---|---|---|---|
| | $AP_{50}$ | $AP_{75}$ | $AP_{50}$ | $AP_{75}$ |
| 100% | 70.05 | 38.38 | 72.31 | 39.49 |
| 30% | 55.73 | 20.14 | 61.25 | 27.91 |
| 10% | 37.71 | 6.98 | 44.61 | 14.97 |

**Branch multiplexing:** An additional experiment that randomly selects from 5% flip or 95% rotation in only one branch (the proportion based on $\lambda = 0.05$ in Table 6) shows $AP_{50}/AP_{75}$: 72.24%/39.51% (DOTA w/o MS) while reducing the training time and the RAM usage to H2RBox's level.

**Padding strategies:** Compared to the performance loss of more than 10% for H2RBox without reflection padding, Table 8 shows that H2RBox-v2 is less sensitive to black borders.

**Annotation noise:** Table 9 multiplies the height and width of annotated HBox by a noise from the uniform distribution $(1 - \sigma, 1 + \sigma)$. When $\sigma = 30\%$, the $AP_{50}$ of H2RBox-v2 drops by only 1.2%, less than H2RBox (2.69%), which demonstrates the better robustness of our method.

**Training data volume:** Table 10 displays that the gap between H2RBox and H2RBox-v2 becomes larger on the sampled version of DOTA, where $p$ denotes the sampling percentage.

## 5    Conclusion

This paper presents H2RBox-v2, a weakly-supervised detector that learns the RBox from the HBox annotation. Unlike the previous version H2RBox, H2RBox-v2 learns the angle directly from the image of the objects through a powerful symmetry-aware self-supervised branch, which further bridges the gap between HBox-supervised and RBox-supervised oriented object detection.

Extensive experiments are then carried out with the following observations: **1)** Compared to H2RBox, H2RBox-v2 achieves higher accuracy on various datasets, with an improvement of 2.32% on average over three versions of DOTA, and 6.33% on the FAIR1M dataset. **2)** H2RBox-v2 is less susceptible to low annotation quality and insufficient training data. As a result, it is compatible with small datasets such as HRSC, which H2RBox cannot handle. **3)** Even compared to fully-supervised counterpart (i.e. Rotated FCOS), H2RBox-v2 still shows quite competitive performance, proving the effectiveness and the potential of symmetry-aware self-supervision for rotating detection.

**Broader impacts.** Oriented object detection can be used for military purposes e.g. by remote sensing.

**Limitations.** There can be cases when the objects are not symmetric in appearance. This may also hold when the objects are partially occluded even from the top view. Moreover, it becomes more challenging to explore the symmetry in 3-D object rotation detection due to occlusion, which yet has been well explored in [39] for autonomous driving by the H2RBox (v1) method [20].

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
