# OpenReview forum: "H2RBox-v2: Incorporating Symmetry for Boosting Horizontal Box Supervised Oriented Object Detection"
_NeurIPS.cc/2023/Conference — NeurIPS 2023 poster_

### Official Review · Reviewer_Pshp · 2023-06-13

**Soundness:** 3 good
**Presentation:** 3 good
**Contribution:** 3 good
**Rating:** 7
**Confidence:** 4

**Summary:**

This paper proposes to exploit the reflection symmetry as a new supervision to HBox-supervised oriented object detection. Several modifications are made to adapt the new self-supervised (SS) branch, including removing the angle subnet in the weakly-supervised (WS) branch and a CircumIoU loss for box regression. Experiments show a clear state-of-the-art result to previous Hbox-supervised oriented object detectors.

**Strengths:**

1. The paper tries to exploit the reflection symmetry to improve the Hbox-supervised object detector, which is interesting.

2. The paper is well-motivated and easy to understand.

3. The proposed H2Rbox-v2 achieves state-of-the-art performance in the Hbox-supervised oriented object detectors.

**Weaknesses:**

Major concern:

1. The soundness of using reflection symmetry is somewhat low. According to page 5 line 130, if the network $f_{nn}(\cdot)$ subjects to both flip and rotate consistencies, then $f_{nn}(I_0)=\theta_{sym}$. I don't know why this equation holds. Clearly, we don't know the symmetrical axis of the image $I_0$. And if we rotate the image by a random angle $\theta$ in the Way2 of Fig.3, then $f_{nn}(I_2)$ would never be equal to $\theta_{sym}$. Even if the symmetrical axis is known, this reflection symmetry learning can only theoretically and ideally work under the setting of a single symmetric object. As a matter of fact, it would be more rational that the converse proposition holds - "If the network $f_{nn}(\cdot)$ can always predict the symmetrical axis $\theta_{sym}$ of the image, then, it must subject to both flip and rotate consistencies." I suggest the authors reconsider the logic here.

2. Due to weakness 1, the novelty is also limited since one can easily conclude that H2Rbox-v2 is to add one more additional branch with a new view generation upon H2Rbox, i.e., the left column of Fig. 2(b) ("vertical flipping"). The right column is random rotation which is similar to the SS branch of H2Rbox. This is what I believe the model brings us performance gains, but not the so-called reflection symmetry learning.  The experiments cannot support that the reflection symmetry is important, but can only verify the effectiveness of adding the vertical flipping branch, which is supported by Tab. 6. It is still unclear for the relationship between reflection symmetry learning and multi-branch self-supervised learning.

3. The ablation studies provide a weak explanation for the effectiveness of the proposed components. In Tab. 4, adding the $l_s$ loss only produces near zero mAP on DOTA, while it is much higher on HRSC. Why did such drastic fluctuations occur? The PSC coder is just an angle encoder-decoder module, which is originally proposed to improve the performance of oriented object detectors. It would not be a critical factor affecting the performance, as demonstrated by the PSC paper, ~2 mAP at most. This paper shows PSC is crucial in the proposed method. That is to say, without PSC, the proposed method produces much poor performance. This largely weakens the soundness of the designs in the SS branch. Additionally, PSC imposes the loss on two encoded phase-shifting patterns of angles, which inherently solves the boundary discontinuity, while the proposed SS branch decodes the phase-shifting patterns to angle and then calculates the loss on two angles, which introduces the boundary discontinuity again, thus undermining the contribution of the snap loss.

4. It is confusing to me that Fig.2 shows the angle is predicted by the SS branch, while all the other properties are predicted by the WS branch. That is to say, the inference time of H2Rbox-v2 (WS+SS) should be double of the WS branch, while H2Rbox only needs the WS branch for inference. I'm concerned about why the FPS of H2Rbox-v2 is equal to H2Rbox as shown in Tab. 2. The authors need to describe the inference process clearly.

Minor concern:

The baseline model is unclear in the ablation studies. In Tab. 4, if the baseline model is "w/o PSC and $l_s$", then what is the angle coder used in the baseline? And what are the angles in calculating Smooth-L1 loss? The same question emerges in Tab. 5 and Tab. 6. Is that mean all the baseline models of the tables adopt the optimal strategies of the other tables? If so, then the best results of Tab.6 seem to be not in line with the others.

Overall:

While the proposed method achieves state-of-the-art performance, my initial rating score is 4. I have some concerns about the soundness of the reflection symmetry learning and the ablation studies. It is of vital importance to give an explanation and analysis of the rationality of the proposed method.

**Questions:**

Suggestions:

1. For the weaknesses 1 and 2, I suggest the author to rethink the logic of the analysis of Sec. 3.1. It would be better if we could restate the analysis in page 5 line 130 to the converse proposition - "If the network $f_{nn}(\cdot)$ can always predict the symmetrical axis $\theta_{sym}$ of the image, then, it must subject to both flip and rotate consistencies." And then the contrapositive - "if the network subjects to neither flip consistency nor rotate consistency, it cannot predict the symmetrical axis of the image." Therefore, we need to enhance both flip consistency and rotate consistency. H2Rbox considers the rotate consistency only, which leads to sub-optimal results.

2. There is a missing experiment, i.e., $\lambda=0$ in Tab. 6.

**Limitations:**

see Weaknesses

---

> ### Author Rebuttal · Authors · 2023-08-09
>
> # To Reviewer Pshp
>
> We sincerely appreciate your valuable suggestions. We hope our clarification could help to make a more informed rating to our work.
>
> **Q1 It would be more rational that the converse proposition holds for symmetric axis prediction and flip/rotate consistency**
>
> We think our (empirically verified) idea is beyond this converse proposition. We restate the main facts and our theory in H2RBox-v2 as:
>
> - **Observation (facts):** Collect a set of images containing a symmetric object along an arbitrary angle, and use this set to train a three-branch like neural network supervised just by the flip and rotate consistencies respectively via the two of the branches **(without the supervision of any annotation)**. We show that the trained network is able to estimate the angle of the symmetric axis of the object.
>
> - **Underlying theory:** We have proved in our paper (see Sec 3.1) that if a function always satisfies the flip and rotate consistencies for the input symmetric image, then the function's output (by certain derivation - see details in the paper) is exactly the angle of the axis of the symmetric image.
>
> - **Handling multiple objects in images:** The above theory strictly holds in the single object case. With an assigner in the SS branch to match objects in different views (see Line 151), the consistency loss is calculated between these matches, so that our "single-object" theory can be applied to each matched object.
>
> **Q2.a One more branch is the key to performance improvement rather than reflection symmetry learning**
>
> - **V2 has completely different angle information acquisition mechanism from v1:** Refer to the general response.
>
> - **Performance improvement is not due to one more branch:** A new experiment that randomly selects from 5% flip or 95% rotation in only one branch (based on $\lambda=0.05$ in Table 6) shows that the number of branches is not the key to improving performance, AP50/AP75:
>
> | Methods | original (2 branches) | multiplex (1 branch) |
> |:-:|:-:|:-:|
> | H2RBox-v2 | 72.31/39.49 | 72.24/39.51 |
>
> **Q2.b Experiments cannot support reflection symmetry is important**
>
> Sorry for leaving an impression of "adding flip to H2RBox-v1" which is in fact not true. Essentially we discard the use of geometric constraints from human annotations in obtaining angles (the way of v1), and the angles are fully learned by the reflection symmetry of objects (refer to the answer of Q2.a). We are not proving that symmetry improves -v1, yet rather that symmetry can replace the way of learning angle in v1.
>
> **Q2.c Relation between reflection symmetry learning and multi-branch SSL**
>
> The former is our theory that allows the network to perceive the angle of objects from their symmetry and the latter is a neural network implementation.
>
> **Q3.a The loss (Tab. 4) produces near 0 mAP on DOTA but much higher on HRSC. Why did such drastic fluctuations occur?**
>
> Without handling boundary discontinuity, we empirically found that the loss could fluctuate in a wide range, even failure in convergence. In comparison, when both PSC and snap loss are used, the training is very stable. The "drastic fluctuations" exactly show the instability and prove the necessity of PSC and snap loss. We now give more discussion in the new version.
>
> **Q3.b Poor performance w/o PSC: PSC would not be a critical factor, ~2 mAP at most**
>
> We agree that the impact of boundary problems may be ~2 mAP in **supervised** setting. Yet the impact of boundary problem in our consistency-based self-supervised setting could be much greater than that in the supervised case, especially in terms of stability, as shown in Table 4. Our point is that, "the poor performance without PSC" suggests in fact H2RBox-v2 allows PSC to well address the boundary problem to avoid poor performance. PSC is indispensable in our approach - but it does not mean our other parts' design is not sound.
>
> **Q3.c Calculating the loss on two decoded angles introduces the boundary discontinuity again**
>
> The snap loss limits the difference between two angles into the $\pi/2$ (see Fig. 4a), thus the calculation won't introduce the boundary discontinuity again. This is why both PSC and snap loss are important to solve boundary problem in the self-supervised design.
>
> **Q4 H2Rbox-v2 predicts angle by the SS branch, while H2Rbox only needs the WS branch for inference. Why is the FPS equal?**
>
> As shown in the WS branch in Fig. 2, due to parameter sharing, the inference is in the form of WS + Angle head from SS, so the efficiency is almost the same as that of H2Rbox.
>
> To be precise, the only additional cost of v2 during inference is PSC decoding. Thus, FCOS/H2RBox-v1/H2RBox-v2 have similar inference times. With input shape (3, 1024, 1024), the accurate costs are:
>
> | | H2RBox | H2RBox-v2 |
> |:-:|:-:|:-:|
> | Flops | 206.91 GFLOPs | 207.01 GFLOPs |
> | Params | 31.92 M | 31.93 M |
>
> **Q5.a When "w/o PSC and $l_s$", what is used in the baseline?**
>
> "w/o PSC" means that the conv layer directly outputs the angle. "w/o $l_s$" means using smooth-L1 loss. We will add the description in our new version.
>
> **Q5.b Do all the tables adopt the optimal strategies of the other tables? Tab.6 seems to be not in line with the others**
>
> Yes, we adopt the optimal strategies if not otherwise specified. The best results are not in line with the others because we run the same config ($\lambda=0.05$) more than once, i.e. DOTA: 40.39/72.59.39.18 (Tab. 6) vs. 40.69/72.31/39.49 (other Tabs.), HRSC: 56.76/89.63/62.93 (Tab. 6) vs. 58.03/89.66/64.8 (other Tabs.), and we forgot to modify the Tab. 6 to the final adopted result. We will unify this result in the final version.
>
> **Q5.c Missing $\lambda=0$ in Tab. 6**
>
> The experiment is now added, which shows that the flip branch is necessary (AP50/AP75/AP):
>
> | $\lambda$ | HRSC | DOTA |
> |:-:|:-:|:-:|
> | 0 | 0.32/0.00/0.06 | 66.37/25.03/31.60 |
>
> Please let us know if there are further questions.

---

> > ### Comment · Reviewer_Pshp · 2023-08-13
> > **Response to authors**
> >
> > Thanks for the detailed rebuttal. Most of my concerns are addressed. The main paper should include the experiment of Q2.a. The authors should add analyses on Q3.a, Q3.B, and Q3.c, and fix the issue of Q5.b. I'm willing to improve my rating score from 4 to 7.

---

> > > ### Author Response · Authors · 2023-08-13
> > >
> > > We will carefully fix these issues in preparing the final version. Thanks again for your nice suggestions and efforts on this paper!

---

### Official Review · Reviewer_tfHb · 2023-06-15

**Soundness:** 3 good
**Presentation:** 2 fair
**Contribution:** 3 good
**Rating:** 5
**Confidence:** 5

**Summary:**

This paper proposes an advance solution for using horizontal bounding boxes as supervision to learn oriented object detectors.
The proposed method H2RBox-V2, which is a modification of the recent work H2RBox, has some technique novelty and contribution as it jointly uses the weakly- and self-supervised branches to learn the rotation angle.
Experiments are validated on DOTA, HRSC and FAIR1M, and show state-of-the-art performance.


**Strengths:**

+ The overall idea to learn the rotated angles from both weakly- and self- supervised branch is very interesting.

+ The task itself, to learn rotated bounding boxes from horizontal bounding boxes, is a new setting for the rotated object detection community.

+ The performance improvement against prior arts is significant.


**Weaknesses:**


- Unclear methodology design and presentation.

(1) The authors claim to propose a new CircumIoU loss for this framework. Unfortunately, in the methodology section, especially Sec3.4, there is no term ‘CircumIoU loss’, and it is impossible to know which loss is the claimed novelty.

(2) Besides, given this unclear description, it is very difficult to know its difference or novelty against some prior works such as [a], which also optimizes the detector from rotation angles.

[a] Arbitrary-Oriented Object Detection with Circular Smooth Label. ECCV 2020.

Other minor issues and comments for improvement:

- Please provide some rotation angle distribution visualizations from the proposed framework.

- Fig.1, column2 and column4. The predictions from the proposed method and SAM-RBox seem not to have much difference. Please consider use more representative figures.

- Table 1, the performance gap. Please explictly mention which dataset it is that leads to 3.41% drop and 0.07% improvement.

- Fig.2 looks very crowded. It can be better polished, and the size of each content can be made more fit.

- For loss function, please use \mathcal{} to distinguish it from scalars.



**Questions:**


The authors have addressed my questions in the rebuttal well.


______ before rebuttal _____________


Q1: Issues on the flaw of the technique framework. Both experimentally and theoretically.

Q2: The details, design and clear presentation of the CircumIoU loss.

Q3: systematically address the unfair experimental comparison issue.


**Limitations:**

The authors have addressed my concerns well.

______________________before rebuttal ___________
The limitations are not properly discussed. The reviewer believes the technique flaws as mentioned in weakness part is more critical to the proposed framework.

---

> ### Author Rebuttal · Authors · 2023-08-09
>
> # To Reviewer tfHb
>
> Thank you for the time and nice suggestions. We humbly point out that there may exist misunderstandings in the review, and we hope our clarification could help on a more informed rating to our work.
>
> **Q1.a Eq. 2 and Eq. 4 conflict: When k is an odd number, the nets learn the opposite rotation angle, i.e. $\theta$ and $\theta + \pi$**
>
> Unfortunately that we didn't well explain or emphasize the basic concept in oriented object detection that bounding boxes have the periodicity of $\pi$.
>
> Angle $\theta$ and $\theta + \pi$ refer to the same bounding box. Similarly, symmetric axis of $\theta$ is also equivalent to symmetric axis of $\theta + \pi$. We will clarify it in our new version.
>
> Based on our above clarification, Eq. 4 is the extended version of Eq. 2, which takes into consideration the equivalence between symmetric axes of $\theta + k\pi$.
>
> **Q1.b Lack experiment: Can using the weakly-supervised branch to predict warrant the same rotation angle as the semi-supervised branch, or at least very close?**
>
> We hope our above explanation could have eased your concerns and potential misunderstanding.
>
> For your specific question here, strictly speaking, our method doesn't have a "semi-supervised branch", do you mean the self-supervised branch? Actually, there is only one angle prediction during inference. If this misunderstanding is aroused by the multiple networks drawn in Fig. 2, we clarify that these networks/branches share the same parameters (see Line 149 and in Fig. 2). Such a parameter-shared graph is widely used (e.g. also in H2RBox). Therefore, no matter which branch is used for inference, the result should be the same in theory.
>
> We further use HRSC to compare the AP50 by using different branches for inference:
>
> | WS | SS-Flip | SS-Rotate |
> |:-:|:-:|:-:|
> | 89.66 | 89.66 | 89.66 |
>
> The above results confirm that with the same input image, all parameter-shared branches have the same output.
>
> **Q2.a The authors claim to propose a new CircumIoU loss, but in the methodology, especially Sec 3.4, there is no term "CircumIoU loss"**
>
> Thanks for your careful review and suggestion. The term "CircumIoU loss" in Sec 3.4 is at Line 182 and in Fig. 4 (b). In Fig. 4 (b), we use a graph similar to the well-known paper "IoU loss" to illustrate the calculation process.
>
> We give a more specific description as follows, which will be added to our new version:
>
> To calculate CircumIoU loss, the predicted box $B_\text{pred}$ is first projected to the direction of ground-truth box $B_\text{gt}$. The obtained projected box $B_\text{proj}$ is displayed as the dashed box in Fig. 4 (b). Afterward, the loss can be calculated as:
>
> $-\ln \frac{intersection(B_\text{proj}, B_\text{gt})}{union(B_\text{proj}, B_\text{gt})}$
>
> CircumIoU loss can enable H2RBox-v2 to use random rotation (RR) data augmentation to further improve the performance (as shown in Tab. 2), which is not supported by H2RBox-v1.
>
> **Q2.b Difference or novelty against prior works such as Arbitrary-Oriented Object Detection with Circular Smooth Label (ECCV 2020)**
>
> Circular Smooth Label (CSL) is about **supervised** learning that learns angle classification from **angle annotations**. However, as described in our paper's title and introduction, H2RBox-v2 is aimed at a different and more challenging setting -- **self-learning** oriented boxes from **horizontal box annotation** without human labeled angle. CSL cannot even work without labeled angles. Therefore, the difference (novelty) between them is huge.
>
> **Q3.a Prior work H2RBox with 1x schedule and multi-scale (MS) has mAP of 74.40%. But in this submission, H2RBox is made deliberately low by removing its MS scheme. More importantly, all the H2RBox-v2 performance is reported with MS scheme. If consider MS for H2RBox, it can even outperform v2**
>
> Thanks for your careful thoughts on our work and giving us the chance to clarify and improve the unclear part of the paper.
>
> Please note that in fact all the H2RBox-v2 performances in this paper are reported with the **single-scale** scheme unless otherwise specified in Table 2 in the paper, where it shows the performance of **H2RBox-v2 using multi-scale: 77.97%**. Compared to the performance of H2RBox with MS reported in the original paper (74.40%), the improvement is even higher than the value 2.26% that we claimed in our paper (see line 229: H2RBox-v2 outperforms H2RBox by 2.26% = 72.31% - 70.05%).
>
> We will add the following comparison (i.e. enrich/update Table 2). Instead of "74.40%", we will use our reproduced result 75.35% (also a higher baseline).
>
> | Methods | 1x w/o MS | 1x w/ MS |
> |:-:|:-:|:-:|
> | H2RBox (R50) | 70.05% | 75.35% |
> | H2RBox-v2 (R50) | 72.31% | 77.97% |
> | Improvement of v2 | +2.26% | +2.62% |
>
> **Q3.b The highest result of H2RBox-v2 79.75% mAP is made by the Swin-transformer. It is more meaningful to also report the H2RBox (in a fair way) with Swin**
>
> Thanks for your suggestion. The new results are as follows (4 GPUs and batchsize=4 to speed up experiments in the limited rebuttal period):
>
> | Methods | 1x w/ MS |
> |:-:|:-:|
> | H2RBox (Swin-Tiny) | 61.60% |
> | H2RBox-v2 (Swin-Tiny) | 79.39% |
> | Improvement of v2 | +17.79% |
> |||
> | H2RBox (Swin-Base) | 61.05% |
> | H2RBox-v2 (Swin-Base) | 80.35% |
> | Improvement of v2 | +19.30% |
>
> On Swin-transformer, -v2 outperforms -v1 by a large margin. But since the original H2RBox paper does not report performance on Swin, it may be doubted that we "made low" the performance of "H2RBox (Swin)" for comparison. Therefore, we contacted the authors of H2RBox about the experiment of H2RBox+Swin. They told us that their experimental results are consistent with ours, but they have not found the reason for this phenomenon so far. Anyway, this result demonstrates H2RBox-v2 has more robustness.
>
> **Q4 Other minor issues**
>
> We will carefully follow these suggestions and revise the paper accordingly. Thanks again for your efforts on this paper!
>
> Please let us know if there are further questions.

---

> > ### Comment · Reviewer_tfHb · 2023-08-10
> > **Response to author rebuttal**
> >
> > Many thanks for the time and effort from the authors.
> >
> > Most of my concerns have been properly addressed, and I realized that the orignal rating before rebuttal is down-graded.
> > Especially, the clarification of significant improvement against H2RBox under *the same and fair setting* makes this work much better than the previous grade.
> >
> > However, before the discussion period ends, still some minor concerns regarding **Q1.b** :
> >
> > - Indeed the authors have clarified that both backbones are weight-sharing before angle prediction. Does it involve any additional layers or parameters to map the frozen backbone feature to the angle?
> >
> > - If the answer of first question is true, then, empercially extract the frozen features and learns the mapping between angle and backbone through the other branch may have some unforseen impact.
> >
> > - In this regard, it would be better for the authors to show some empercial outcomes on this aspect, so that my last remaining concern is resolved.
> >
> > I am willing to improve my rating to the accept threshold if these final concerns can be properly addressed in the following week.

---

> > > ### Author Response · Authors · 2023-08-11
> > >
> > > Thanks for your prompt and responsible response. We try to get your point and our tentative response is as follows.
> > >
> > > **Q1: Does it involve any additional layers or parameters to map the frozen backbone feature to the angle?**
> > >
> > > Following backbone, there is an angle head, consisting of several convolution layers and a PSC decoder, to map the feature to the angle (see Fig.2b). The angle heads in different branches also share the same parameters. In fact, all heads, including regression, classification, center-ness and angle, their parameters are also shared, which is consistent with the classic detectors, i.e. RetinaNet and FCOS.
> > >
> > > **Q2: Extract the frozen features and learn the mapping between angle and backbone through the other branch may have some unforeseen impact**
> > >
> > > Please note that our backbone is not frozen **during training** -- the backbone, angle head, and other heads (i.e. regression, classification, center-ness) are updated together by SS and WS losses. Once the training is complete, both the backbone and the heads are frozen, and there is no need to generate SS views or learn the angle mapping **during inference**.
> > >
> > > In oriented object detection, it is common to predict angle in a separate head/branch, e.g. in your mentioned prior work CSL, the angle is also separately predicted by an angle head which is trained by an independent angle loss. The detection results and the performance show that our design works well.
> > >
> > > Please let us know if we misunderstood your questions, and also let us know if there are further questions.

---

> > > > ### Comment · Reviewer_tfHb · 2023-08-11
> > > > **Response to author rebuttal (Round 2)**
> > > >
> > > > Many thanks for the authors to provide informative and timely response.
> > > >
> > > > - Regarding **Q1**, as all the parameters are all shared, then my concerns have been well addressed.
> > > >
> > > > - Regrading **Q2**, yes, I think the reviewers have clearly catched my point. Although there is no emperically justification, I think the explanation is responable.
> > > >
> > > > After the rebuttal and the clarification, I think now I hold more positive than negative view on this work. I will revise my comments and raise my score in the following day, as it takes some time.
> > > >
> > > > Meanwhile, I would kindly suggest the authors:
> > > >
> > > > - if possible, please revise Fig.2 in the camera-ready version in a more clear way.
> > > >
> > > > - if possible, CircumIoU loss part can be presented in a more informative way in the camera-ready version.
> > > >
> > > > Finally, I would like to thank the authors again for their patient and clear response.

---

> > > > > ### Author Response · Authors · 2023-08-13
> > > > >
> > > > > We will carefully revise the paper and fix these issues. Thanks again for your recognition and valuable suggestions on this paper!

---

### Official Review · Reviewer_W3Ay · 2023-07-04

**Soundness:** 3 good
**Presentation:** 3 good
**Contribution:** 3 good
**Rating:** 7
**Confidence:** 5

**Summary:**

This paper introduces H2RBox-v2, an innovative approach to further bridge the gap between HBox-supervised and RBox-supervised oriented object detection. It seeks to address the limitations of the original H2RBox model, which required high-quality annotations and large training datasets, and was incompatible with rotation augmentation. H2RBox-v2 augments the original model by adding a self-supervised branch that learns object orientations from inherent visual symmetry, and a weakly-supervised branch that incorporates a new CircumIoU loss to allow for random rotation augmentation.

**Strengths:**

- **Originality:** While the paper builds upon existing work (specifically H2RBox), it infuses new concepts, the most significant being the utilization of symmetry for angle regression. This concept is highly innovative and introduces a novel angle of approach for oriented object detection. This creativity in applying the natural property of symmetry to enhance detection accuracy sets this work apart from previous methods and broadens the boundary of the field.

- **Quality:** The proposed H2RBox-v2 exhibits enhanced performance on various datasets compared to its predecessor, extensive ablation experiments have demonstrated the importance of each module.

- **Clarity:** The paper is well-structured and clearly presents the methodology and results, making the contributions of this research easily understandable.

- **Significance:** Bridging the gap between HBox-supervised and RBox-supervised oriented object detection significantly reduce the annotation costs.

- H2RBox-v2 has shown improved performance on various datasets and is specifically designed to cope with situations where H2RBox-v1 may underperform. This makes it a valuable contribution to real-world applications of oriented object detection.

**Weaknesses:**

- **Theory is not fully comprehensive:** *L138-142* only discussed the case where a single instance is contained in an image. However, in reality, remote sensing images can contain many objects, especially in dense scenarios. Despite the theoretical proof being not fully comprehensive, their simplicity of idea and empirical effectiveness seem to be sufficient according to me.

- **Training overhead not detailed:** It is crucial that the model does not introduce additional overhead during the testing phase, but the paper does not discuss the computational overhead and the time taken for the model training process in detail

**Questions:**

- Would it be more effective to conduct the comparative experiments on the DOTA dataset, regarding the assertion that H2Rbox-v1 requires more training data than v2? Specifically, by training both v1 and v2 with varying percentages of DOTA data (namely 10%, 20%, and 50%), we could gain a more robust comparison of their respective performances.

- When the object is not in the center of the image, are Way1 and Way2 described in L124-132 still equivalent?

- In remote sensing scenarios, some objects may not possess symmetry, such as swimming pools and harbors. What impact does this have on the theory?

**Limitations:**

Nothing to report.

---

> ### Author Rebuttal · Authors · 2023-08-09
>
> # To Reviewer W3Ay
>
> Thanks for your positive comments and constructive suggestions. Your endorsement of our method gives us significant encouragement.
>
> **Q1 The computational overhead and the time taken for the model training/testing process**
>
> **Test phase:** In our submission, we only gave FPS for evaluating inference time (in Table 2). We now add the accurate computational costs (the additional cost of -v2 is due to PSC decoding) and we will also enrich the experiment section in our new version:
>
> |        |  H2RBox (v1)  |   H2RBox-v2   |
> |:------:|:-------------:|:-------------:|
> | Flops  | 206.91 GFLOPs | 207.01 GFLOPs |
> | Params |    31.92 M    |    31.93 M    |
>
> **Train phase:** H2RBox-v2 is slower than -v1 in training as it involves one more branch. Here we provide an additional experiment that randomly selects from 5% flip or 95% rotation in only one branch ("5%" is based on $\lambda=0.05$ in Table 6). The resulting AP50/AP75 are:
>
> |  Methods  | original (2 branches) | multiplex (1 branch) |
> |:---------:|:---------------------:|:--------------------:|
> | H2RBox-v2 |      72.31/39.49      |     72.24/39.51      |
>
> The multiplex version requires similar training time to -v1 while keeping high performance as -v2:
>
> | FCOS  | H2RBox-v1 | H2RBox-v2 | H2RBox-v2 (multiplex) |
> |:-----:|:---------:|:---------:|:---------------------:|
> | 5h10m |   7h10m   |   8h56m   |         7h7m          |
>
> **Q2 Comparative experiments on the varying percentages of DOTA dataset, regarding the assertion that H2Rbox-v1 requires more training data than v2**
>
> Thanks for your nice suggestions. In the following table, we show that the gap between -v1 and -v2 becomes larger on the sampled version of DOTA dataset (30%, 10%). The AP50/AP75 are:
>
> |  Methods  |     Full    |     30%     |     10%     |
> |:---------:|:-----------:|:-----------:|:-----------:|
> | H2RBox    | 70.05/38.38 | 55.73/20.14 | 37.71/ 6.98 |
> | H2RBox-v2 | 72.31/39.49 | 61.25/27.91 | 44.61/14.97 |
>
> **Q3 When the object is not in the center of the image, are Way1 and Way2 still equivalent?**
>
> Yes, actually for any image and any $\theta$, { flip about line $\theta$ } is equivalent to { flip vertically and then rotate by $2 \theta$ }. But when the object is not in the center, the input image becomes asymmetric, so $f_\text{nn}\left ( I' \right ) = f_\text{nn}\left ( I_0 \right )$ (on Line 125) does not hold.
>
> Technically speaking, when the object is not in the center, the situation is similar to multiple object detection. Although our theoretical study is performed on a single object, there is an assigner in the SS branch to match the center of objects in different views (see Line 151), and the consistency loss is calculated between these matched center points. With the assigner, our "single-object" theory can be applied to each matched object center, and this is the way our network can be used for multiple objects (and objects not in the center).
>
> **Q4 Some objects may not possess symmetry, e.g. swimming pools and harbors. What impact does this have on the theory?**
>
> While the training objects are preferred to be symmetric which often holds in aerial images, our experiments show that the symmetry need not be strictly obeyed. H2RBox-v2 can still optimize for the most likely solution -- an approximate axis that divides the object into two "most mirrored" parts. This mechanism extends the applicability of H2RBox-v2 to most elongated objects. As a result, the performance of each class and the visualization in our supplementary material demonstrates that H2RBox-v2 still gives a competitive performance in terms of swimming pools and harbors.
>
> Please let us know if there are further questions.

---

> > ### Comment · Reviewer_W3Ay · 2023-08-16
> >
> > I would thank the authors for addressing my concerns. Given my current rating of 7, I intend to maintain it, unless other reviewers introduce new issues that warrant reconsideration.

---

> > > ### Author Response · Authors · 2023-08-16
> > >
> > > We will carefully prepare the final version. Thanks again for your recognition and valuable suggestions!

---

### Official Review · Reviewer_k1cY · 2023-07-05

**Soundness:** 3 good
**Presentation:** 4 excellent
**Contribution:** 3 good
**Rating:** 7
**Confidence:** 5

**Summary:**

This paper proposes a new horizontal box-supervised rotation object detection detector. The proposed detector consists of two modules: a self-supervised regression branch for angle regression and a weakly supervised branch for horizontal box regression. This method is more simpler and shows clear improvement over the previous one.

**Strengths:**

- The paper fully considers the independence and correlation between angle and horizontal box in rotation object detection. It only uses angle regression in self-supervised regression while using Circumscribed RBox IoU to associate angle regression and horizontal box regression in weakly supervised regression to obtain the rotated box.
-  The paper greatly improves the performance of the hbox-supervised detector, approaching or even reaching the performance of some rbox-supervised detectors.
- The paper is well written, and the method is clear and well described.

**Weaknesses:**

- Figure 2 seems a bit complicated, it would be better to highlight the main points to make it clearer.
- Although this article has achieved impressive results, it is important to consider the robustness of the model when horizontal bounding box annotations are not accurate enough, the article's experiments do not explicitly demonstrate the robustness towards inaccuracies in horizontal bounding box annotations.

**Questions:**

- I have a question about the self-supervised branch in the proposed method. Why was it designed with two perspectives instead of using one perspective and randomly selecting multiple variations under that perspective? Additionally, would using a more perspective, such as different rotation angles, combinations of rotation and symmetries, or scaling variations, lead to performance improvements?

- Regarding Table 4, I noticed that the experiments on DOTA and HRSC datasets show inconsistent results. Specifically, in the Dota dataset, not using PSC will result in an incorrect result, while in the HRSC dataset, not using PSC only results in a performance loss. This raises the question of whether PSC only improves performance or if it has an indispensable impact on the model.

---

> ### Author Rebuttal · Authors · 2023-08-09
>
> # To Review k1cY
>
> Thank you for the nice comments and valuable suggestions. By revising accordingly, the article is now clearer and more complete!
>
> **Q1 Robustness to inaccuracies in horizontal bounding box annotations**
>
> Thanks. We add some random noise to the annotation and record AP50/AP75 under different noise levels on DOTA-v1.0. Noise=30% indecates Height=Height*[0.7-1.3] and Width=Width*[0.7-1.3], both from uniform distribution. The results are as follows:
>
> | Noise     |      0%     |     10%     |     30%     |     50%     |
> |:---------:|:-----------:|:-----------:|:-----------:|:-----------:|
> | H2RBox    | 70.05/38.38 | 69.19/35.24 | 67.39/26.02 | 61.66/14.55 |
> | H2RBox-v2 | 72.31/39.49 | 71.68/36.33 | 71.11/34.12 | 67.88/21.56 |
>
> Results show that when adding 30% random noise to the annotations, the AP50 of H2RBox-v2 drops by only 1.2%, less than H2RBox (2.69%), which demonstrates the better robustness of our method. We will also add these results in the new version to make it more convincing.
>
> **Q2.a About the self-supervised branch**
>
> Thank you for bringing up this interesting idea. Our two-perspective paradigm is intuitively derived from our theory which involves two consistencies, and you provide a new multiplexing solution for symmetry-aware learning. According to your suggestion, we conduct two additional experiments:
>
> 1. Randomly select from 5% flip or 95% rotation in one perspective/branch.
> 2. Use one more perspective/branch with another random rotation angle.
>
> The results (AP50/AP75) are as follows:
>
> | Exp ID    |   original  |rand sel.    | +branch     |
> |:---------:|:-----------:|:-----------:|:-----------:|
> | H2RBox-v2 | 72.31/39.49 | 72.24/39.51 | 72.05/39.18 |
>
> The results show that the multiplexing setting reaches almost the same accuracy as the original one. We will also add this result in the new version.
>
> **Q2.b Using scaling variations**
>
> DOTA usually follows two protocols: with and without multi-scale (MS). In the "w/o MS" setting, using scaling can improve the performance, but it also makes the comparison unfair if we integrate scaling and compare with baselines without MS. Whereas, in the "w/ MS" setting, reflection/scaling are already used as augmentation. In this case, we suppose using them in view generation plays the same role as in augmentation, on the ground that scaling does not change the angle, and the consistency loss between the original view and the scaling one is zero. We will further explore your suggestions in future work.
>
> **Q3 Inconsistent results in Table 4. Does PSC have an indispensable impact?**
>
> PSC is indispensable for the stability of our model.
>
> Without PSC, we had empirically observed that the loss could fluctuate in a wide range (possibly due to the angular boundary discontinuity), even failure in convergence. In comparison, when both PSC and snap loss are used, the training is very stable, with not a single failure in our entire experiments.
>
> In terms of your mentioned results, they seem to be inconsistent, but the instability is consistent, and both results prove the necessity of PSC and snap loss.
>
> Please let us know if there are further questions.

---

> > ### Comment · Reviewer_k1cY · 2023-08-13
> > **My final decision**
> >
> > Thanks for the response. I think it all makes sense, and glad to see the authors added the experiment of using one perspective/branch, leading to an improvement in H2Rbox-v2’s performance compared to v1 without any adverse effects. Thanks for the good paper. Now I do not find any other fatal problems, and I will increase my rating from 6 to 7.

---

> > > ### Author Response · Authors · 2023-08-13
> > >
> > > We will carefully revise and prepare the final version. Thanks again for your nice words and constructive suggestions on this paper!

---

### Author Rebuttal · Authors · 2023-08-09

General Response:

We thank the reviewers for their time and constructive suggestions. And the reviewers give appreication in a few points:

1. writing/presentation (**k1cY**:The paper is well written, and the method is clear and well described; **W3Ay**: The paper is well-structured and clearly presented; **Pshp**: The paper is easy to understand)

2. motivation/methodology (**tfHb**: The idea to learn the angles is very interesting; **Pshp**: Exploiting the reflection symmetry to improve the object detector is interesting and the paper is well-motivated; **W3Ay**: The paper infuses new concepts -- utilizing symmetry for angle regression, which is highly innovative; **k1cY**: The paper fully considers the independence and correlation between angle and horizontal box, and it only learns angles in self-supervised regression)

3. experiments/results (**k1cY**: The paper greatly improves the performance, approaching or even reaching the performance of some rbox-supervised detectors; **W3Ay**: It exhibits enhanced performance on various datasets and extensive ablation experiments have demonstrated the importance of each module; **Pshp**: The proposed H2Rbox-v2 achieves state-of-the-art performance in the Hbox-supervised oriented object detectors)

However, there are also some major concerns as follows, in which we have to humbly suggest that there may exist misunderstandings.

Q1. **Unfair comparison** (**tfHb**: Comparison with prior work H2RBox is unfair. All the H2RBox-v2 performance is reported with the multi-scale scheme, but H2RBox is made deliberately low by removing its multi-scale scheme)

Our response: All the H2RBox-v2 performances are reported with the **single-scale** scheme unless otherwise specified in Table 2 in the paper. In our comparison (see line 229: H2RBox-v2 outperforms H2RBox by 2.26% = 72.31% - 70.05%), both sides are based on single-scale. When comparing on multi-scale scheme (according to the reviewer's suggestion), the improvement is 2.62%, even higher than the value 2.26% that we claimed.

Q2. **Soundness of the theory** (**tfHb**: Eq. 2 and Eq. 4 conflict: When k is an odd number, the nets learn the opposite rotation angle, i.e. $\theta$ and $\theta + \pi$; **Pshp**: I don't know why the equation $f_\text{nn}\left ( I_0 \right ) = \theta_\text{sym}$ holds)

Our response:

**to tfHb:** In oriented object detection, bounding boxes have the periodicity of $\pi$. Angle $\theta$ and $\theta + \pi$ refer to the same bounding box. Similarly, symmetric axis of $\theta$ is also equivalent to symmetric axis of $\theta + \pi$.

**to Pshp:** The equation is solved from $f_\text{nn}\left ( I_0 \right ) = \theta_\text{pred} = - \theta_\text{pred} + 2\theta_\text{sym}$, which is mathematically derived from the equivalence between Way 1 and Way 2 (see definition of the two ways in Line 124-127).

Q3. **Improvement not due to the new theory** (**Pshp**: One more branch than H2RBox (v1) is the key to performance improvement rather than so-called reflection symmetry learning)

Our response: H2RBox-v2 has a completely different angle information acquisition mechanism from v1. To prove v2 learns angle from the image (unlike v1 from annotation), we provide an additional experiment (see the attached PDF in this rebuttal). We weaken the annotation of DOTA to square boxes so that they don't contain any angle-related information. The results show that in this experiment v1 fails to learn the correct angle, while v2 still finds the correct angle. This verifies the soundness of our theory (If our theory doesn't work, v2 should be the same as v1 in this experiment).

We hope our clarification could help to make a more informed rating to our work. In the following individual response, we provide answers to each raised weakness/question.

Best regards,

Authors

---

### Decision · Program_Chairs · 2023-09-21

**Decision:**

Accept (poster)

**Comment:**

The paper proposes an improved version of H2RBox for oriented bounding box detection. It is an interesting but infrequently discussed question in the literature, while all the reviewers are positive about the paper after the rebuttal. The AC recommends accepting the paper.